# Early Life Obesity Increases Neuroinflammation, Amyloid Beta Deposition, and Cognitive Decline in a Mouse Model of Alzheimer’s Disease

**DOI:** 10.3390/nu15112494

**Published:** 2023-05-27

**Authors:** Simon W. So, Kendra M. Fleming, Joshua P. Nixon, Tammy A. Butterick

**Affiliations:** 1Minneapolis Veterans Affairs Health Care System, Minneapolis, MN 55417, USA; soxxx063@umn.edu (S.W.S.); kendra.fleming@va.gov (K.M.F.); nixon049@umn.edu (J.P.N.); 2Department of Neuroscience, University of Minnesota Twin Cities, Minneapolis, MN 55455, USA; 3Department of Food Science and Nutrition, University of Minnesota Twin Cities, St. Paul, MN 55108, USA; 4Department of Surgery, University of Minnesota Medical School, Minneapolis, MN 55455, USA

**Keywords:** obesity, amyloid beta, Alzheimer’s disease, cognitive decline, inflammation

## Abstract

Obesity, a known risk factor of Alzheimer’s disease (AD), increases the activation of microglia, leading to a proinflammatory phenotype. Our previous work shows that a high fat diet (HFD) can cause neuroinflammation and cognitive decline in mice. We hypothesized that proinflammatory activation of brain microglia in obesity exacerbates AD pathology and increases the accumulation of amyloid beta (Aβ) plaques. Presently, we tested cognitive function in 8-month-old male and female APP/PS1 mice fed a HFD, starting at 1.5 months of age. Locomotor activity, anxiety-like behavior, behavioral despair, and spatial memory were all assessed through behavioral tests. Microgliosis and Aβ deposition were measured in multiple brain regions through immunohistochemical analysis. Our results show that a HFD decreases locomotor activity, while increasing anxiety-like behavior and behavioral despair independent of genotype. A HFD led to increased memory deficits in both sexes, with HFD-fed APP/PS1 mice performing the worst out of all groups. Immunohistochemical analysis showed increased microgliosis in mice fed a HFD. This was accompanied by an increase in Aβ deposition in the HFD-fed APP/PS1 mice. Together, our results support that HFD-induced obesity exacerbates neuroinflammation and Aβ deposition in a young adult AD mouse model, leading to increased memory deficits and cognitive decline in both sexes.

## 1. Introduction

Obesity is linked with an earlier onset of Alzheimer’s disease (AD) and increased AD neuropathology [1,2]. Studies have shown a correlation between increased early adulthood body mass index (BMI) and late-life dementia in both sexes [3]. Obesity is also associated with increased dietary fat and saturated fat intake [4,5]. Diets high in saturated fatty acids are known to potentiate neuroinflammatory diseases [6,7,8]. In microglia, the immune cells of the brain, exposure to dietary saturated fatty acids triggers a shift towards active inflammatory phenotypes. We have previously shown that a high fat diet (HFD) leads to increased hippocampal microgliosis and cognitive decline in mice [9,10]. As neuroinflammation and microglial activation are factors implicated in the etiology of AD [11], we hypothesize that proinflammatory activation of brain microglia in obese young adult mice exacerbates AD pathology and increases accumulation of the amyloid beta (Aβ) plaques characteristic of AD [6].

The mPrPAPPSwe/PS1ΔE9 (APP/PS1) transgenic mouse model of AD has provided an increased understanding of progressive Aβ-dependent neurodegeneration [12]. This model overexpresses the human APP transgene and leads to the development of Aβ plaques by 4–6 months, and spatial learning and memory deficits starting at 7 months of age [13,14,15]. As in AD, the APP/PS1 model develops severe astroglial and microglial reactions with the onset of amyloid pathology [12,13]. Thus, the model provides a robust platform to study the relationship between HFD, microglial metabolic adaptation, and Aβ pathology. Previous studies involving this model have included starting the diet during mature adulthood and observing behavior and disease pathology during late adulthood [16,17]. As early adulthood obesity has been correlated with late-life dementia independent of mid- and late-life obesity [3], we examined the effects of early long-term HFD feeding in this model. Other early-HFD studies using this model have focused on the use of only male mice [18]. We included both male and female mice, as studies have shown that women have a near twofold increase in risk of developing Alzheimer’s disease compared to men [19]. In the present study, male and female APP/PS1 and wild type (WT) mice were fed either 60% HFD or normal chow (NC) for 6 months, starting at 1.5 months of age. In both sexes, Barnes maze and Y-maze tests showed that HFD-fed APP/PS1 mice had an increase in memory impairment compared to HFD-fed WT mice. In females, HFD-fed APP/PS1 and WT mice had decreased locomotor activity, increased anxiety-like behavior, and increased behavioral despair compared to NC-fed mice. Immunohistochemical analysis of IBA1 (ionized calcium binding adaptor molecule 1) and Aβ deposits (Aβ12F4 and Aβ4G8) showed that HFD-fed APP/PS1 mice had an increase in microgliosis and Aβ pathology in multiple brain regions. Together, these results support that HFD exposure during early adulthood exacerbates cognitive decline and AD pathology in male and female APP/PS1 mice.

## 2. Materials and Methods

### 2.1. Animals

The mPrP-APPSwe/PS1ΔE9 (APP/PS1, also referred to as “AD mouse”) mouse model of Alzheimer’s Disease has a modest overexpression of the human APP transgene (~2–3 fold over non-transgenic) [14], which leads to the development of Aβ plaques by 4–6 months, and spatial learning and memory deficits starting at 7 months of age [13,14,15]. Genotyping was performed via a quantitative real-time polymerase chain reaction (qRT-PCR) (Transnetyx, Cordova, TN, USA). 1.5-month-old male and female APP/PS1 and WT mice were fed either 60% HFD or NC for 6 months. Body composition measurements and behavioral tests were performed after the diet period and were followed by termination and tissue collection. Mice were obtained from the breeding colony at the Minneapolis VA Medical Center, and the experimental protocol was approved by the Minneapolis VA Health Care System Institutional Animal Care and Use Committee.

### 2.2. Body Composition

Body composition for all groups (n = 10–14) was measured using an EchoMRI 700 (Echo Medical Systems, Houston, TX, USA) before HFD exposure and at 6 months on assigned diet [20,21]. Changes in fat mass and lean mass were calculated and statistically analyzed using GraphPad Prism 9 software (GraphPad, San Diego, CA, USA).

### 2.3. Open Field Test

An open field test was used to measure locomotor activity and anxiety-like behavior for all groups (n = 12–21). Animals were removed from their home cage and placed individually into a novel arena (50.8 cm × 50.8 cm × 25.4 cm) over a 30 min period. The outer 8.9 cm of the arena was designated as the “outer” region, with the center designated as the “inner” region. Activity was collected via ANY-maze tracking software (ANY-maze V6.3, Wood Dale, IL, USA) synced with a camera. Total distance traveled and average velocity were collected for locomotor activity and time spent in the inner and outer sections were collected for anxiety-like behavior. Anxiety-like behavior was evaluated by measuring the degree of thigmotaxis, or the tendency of the mouse to remain close to walls [22]. Therefore, a lower inner/outer ratio was deemed to be a display of higher anxiety-like behavior. Data were statistically analyzed using GraphPad Prism 9 software.

### 2.4. Porsolt Forced Swimming Test

The Porsolt forced swimming test was used to measure behavioral despair [23] for all groups (n = 13–18). A cylindrical tank with height 30 cm and diameter 20 cm was filled with water at 15 cm deep, kept at 25 °C. Light intensity in the room was kept at 1000 lx and curtains surrounded cylinders in all directions to minimize light and other spatial cues. Mice were placed in the tank and activity was recorded for 6 min. Time spent immobile was collected via ANY-maze tracking software. A mouse was deemed immobile when it was floating 60% passively for at least 1.5 s. After testing, the mouse was dried with a clean towel, placed back into its home cage with bedding, which was placed halfway onto a warming pad for 15 min to allow for thermoregulation. Data were statistically analyzed using GraphPad Prism 9 software.

### 2.5. Y-Maze Test

The Y-Maze spontaneous alternation test was used to examine working spatial memory [24] for all groups (n = 10–14). The maze consisted of 3 equally spaced arms that were 120 degrees apart and of identical dimensions. Distal cues were visible from the maze. The mice explored the maze for 8 min and arm entries were tracked using ANY-maze software. Spontaneous alternation percentage was calculated by taking the number of consecutive entries into 3 different arms, divided by the total number of alternations (which also equaled total entries minus 2). Data were statistically analyzed using GraphPad Prism 9 software.

### 2.6. Barnes Maze Test

Spatial memory was tested using the Barnes maze as previously described [25] for all groups (n = 11–17). Mice were placed on a white circular maze (91.44 cm diameter) consisting of 20 evenly spaced holes (7.5 cm apart, 5 cm diameter) located 2 cm from the perimeter. The target hole, a black box under one of the holes, remained in the same place during the training period. Spatial cues were visible from the maze and were kept constant during the training period. At the time of 30 s before the task, mice were placed in a black chamber at the center of the maze. After removal of the chamber, mice used spatial cues to orient themselves and locate the target hole. The session ended when the mouse completely entered the escape box or after 3 min elapsed. If the mouse did not enter the hole within the allotted time, it was gently guided into the hole. Lights were shut after each session, and the mouse was allowed to stay in the escape box for 10 s before being brought back to its home cage. Mice were trained for a total of 4 training sessions per day with at least a 10 min inter-trial interval for a total of 4 consecutive days. Latency to target hole and average distance from target hole were measured over the 4 training days with ANY-maze software. The probe test was performed on the 5th day to measure spatial memory. The escape hole was removed, and the mouse was allowed to explore the maze for 90 s with the latency to the target hole being recorded [25,26,27,28]. Data were statistically analyzed using GraphPad Prism 9 software.

### 2.7. Blood Glucose

Non-fasted blood glucose was measured via tail vein puncture prior to termination using a glucometer (MediSense, Bedford, MA, USA) for all groups (n = 6–12). Data were statistically analyzed using GraphPad Prism 9 software.

### 2.8. Histology

Brain tissue for all groups (n = 4–5) were fixed in 10% neutral buffered formalin (Cancer Diagnostics, Durham, NC, USA) for 48 h after collection. Tissue was then put in 20% sucrose (Sigma, Burlington, MA, USA) in phosphate-buffered saline (PBS) (Thermo Scientific, Waltham, MA, USA) solution for 48 h. Brains were then submerged in cryoprotectant (30% sucrose, 30% ethylene glycol (Sigma, Burlington, MA, USA), 1% polyvinylpyrrolidone 40 (Sigma, Burlington, MA, USA), in PBS) at −20 °C until the time for sectioning. Before sectioning, tissue was submerged in 20% sucrose at 4 °C for 24 h.

For 40 μm sections, a sliding microtome was used. Antigen retrieval was carried out for 30 min at 78 °C (IBA1: Reveal Decloaker, Biocare Medical, Pacheco, CA, USA; Aβ12F4 and Aβ4G8: Rodent Decloaker, Biocare Medical, Pacheco, CA, USA). For Aβ12F4 and Aβ4G8 stains, sections were washed in 70% formic acid (Sigma, Burlington, MA, USA) for 10 min. Blocking of peroxidase was performed for 20 min using 3% H_2_O_2_ (Sigma, Burlington, MA, USA). Sections were blocked for 15 min (100% Background Sniper; Biocare Medical, Pacheco, CA, USA). Sections were incubated overnight in primary antibody (IBA1: 1.25 × 10^−5^ mg/mL, Fujifilm Wako, Richmond, VA, USA; Aβ12F4: 1.0 × 10^−4^ mg/mL, Biolegend, San Diego, CA, USA; Aβ4G8: 1.0 × 10^−4^ mg/mL, Biolegend, San Diego, CA, USA) solution at 4 °C. Sections were incubated in secondary antibody (IBA1: goat anti-rabbit, 2.5 × 10^−4^ mg/mL, Jackson ImmunoResearch, West Grove, PA, USA; Aβ12F4 and Aβ4G8: goat anti-mouse, 1.3 × 10^−3^ mg/mL, Jackson ImmunoResearch, West Grove, PA, USA) for 1 h. Artificial peroxidase (Vectastain; Vector Laboratories, Burlingame, CA, USA) was used to block for 1 h. 3,3’-diaminobenzidine (DAB) reaction was conducted for 3 min using 4% chromogen (BioLegend, San Diego, CA, USA) in substrate buffer (BioLegend, San Diego, CA, USA). Sections were then mounted, coverslipped, and scanned at 40× brightfield (Huron TissueScope LE; Huron Digital Pathology, St. Jacobs, ON, Canada). Densitometry of the hippocampus, cortex, and hypothalamus was performed using Fiji software and statistically analyzed using GraphPad Prism 9 software.

For 5 μm sections, histology was performed by the University of Minnesota’s Comparative Pathology Shared Resource (CPSR). Tissues were embedded in paraffin, sectioned, and mounted onto slides. Tissue sections were deparaffinized, rehydrated to water, and subjected to antigen retrieval using heat and citrate buffer. Endogenous peroxidase was blocked using 3% hydrogen peroxide followed by Dako Protein Serum Block (Dako, Santa Clara, CA, USA). For IBA1, the primary antibody (1:600, Biocare Medical, Pacheco, CA, USA) was incubated at room temperature for 1 h. Detection was achieved using Rabbit EnVision™+ Kit (Dako, Santa Clara, CA, USA) and developed using diaminobenzidine chromogen (Dako, Santa Clara, CA, USA). For Aβ4G8, the primary antibody (1:1600, Biolegend, San Diego, CA, USA) was incubated for 1 h at room temperature. Detection was achieved using Vector M.O.M ImmPress kit (Vector Laboratories, Burlingame, CA, USA) and developed using diaminobenzidine chromogen (Dako, Santa Clara, CA, USA). For Aβ12F4, the primary antibody (1:250, Millipore, Burlington, MA, USA) was incubated at room temperature for 1 h. Detection was achieved using Mouse EnVision™+ Kit (Dako, Santa Clara, CA, USA) and developed using diaminobenzidine chromogen (Dako, Santa Clara, CA, USA). All slides were counterstained with Mayer’s Hematoxylin. Slides were scanned at 40× brightfield (Huron TissueScope LE; Huron Digital Pathology, St. Jacobs, ON, Canada). Densitometry of the hippocampus, cortex, and hypothalamus was performed using Fiji software and statistically analyzed using GraphPad Prism 9 software.

## 3. Results

### 3.1. HFD Increases Fat Mass and Lean Mass in Both Sexes

EchoMRI was used to measure body composition. Regardless of sex, there was a significant main effect of diet on fat mass changes (Females: F_1, 39_ = 410.3, *p* < 0.0001; Males: F_1, 25_ = 29.22, *p* < 0.0001) and lean mass changes (Females: F_1, 39_ = 27.06, *p* < 0.0001; Males: F_1, 28_ = 25.72, *p* < 0.0001). There was no significant main effect of genotype on fat mass changes (Females: F_1, 39_ = 0.01, *p* = 0.9101; Males: F_1, 25_ = 0.002, *p* = 0.9605) nor lean mass changes (Females: F_1, 39_ = 0.46, *p* = 0.5015; Males: F_1, 28_ = 2.88, *p* = 0.1010). There was also a significant interaction between diet and genotype on fat mass changes in females but not males (Females: F_1, 39_ = 4.33, *p* = 0.0441; Males: F_1, 25_ = 1.38, *p* = 0.2508) and not for lean mass changes in either sex (Females: F_1, 39_ = 0.21, *p* = 0.6532; Males: F_1, 28_ = 0.39, *p* = 0.5379). In both females and males, post hoc tests showed that HFD groups had a larger increase in both fat mass (Females: *p* < 0.0001, Figure 1A; Males: *p* < 0.05, Figure 1C) and lean mass (Females: *p* < 0.05, Figure 1B; Males: *p* < 0.05, Figure 1D) compared to NC groups.

### 3.2. HFD Decreases Locomotor Activity and Increases Anxiety-like Behavior Compared to NC in Females

An open field test was used to measure locomotor activity and anxiety-like behavior. In females, differences in total distance traveled were attributed to diet (F_1, 69_ = 27.36, *p* < 0.0001) and genotype (F_1, 69_ = 4.773, *p* = 0.0323), with a significant interaction between the two (F_1, 69_ = 6.417, *p* = 0.0136). NC-fed AD mice had increased total distance traveled relative to NC-fed WT (*p* < 0.01, Figure 2A) or to HFD-fed mice regardless of genotype (AD NC *p* < 0.0001 vs. AD HFD or WT HFD, Figure 2A). Diet also impacted average velocity in female mice (F_1, 74_ = 22.17, *p* < 0.0001), but genotype did not (F_1, 74_ = 2.812, *p* = 0.0978) and there was no significant interaction between the two (F_1, 74_ = 3.516, *p* = 0.0647). As with distance traveled, NC-fed AD mice had an overall increase in average velocity compared to all other groups (*p* < 0.05 vs. WT NC; *p* < 0.001 vs. AD HFD; *p* 0.0001 vs. WT HFD; Figure 2B). For anxiety-like behavior, diet (F_1, 73_ = 29.49, *p* < 0.0001) and genotype F_1, 73_ = 4.500, *p* = 0.0373) significantly impacted differences in inner/outer ratio, with no significant interaction between the two (F_1, 73_ = 0.013, *p* = 0.9108). A HFD decreased inner/outer ratio in both WT (*p* < 0.001 vs. WT HFD) and AD mice (*p* < 0.01 vs. AD HFD, Figure 2C).

In males, there was a significant main effect of diet on total distance traveled (F_1, 52_ = 29.10, *p* < 0.0001) and total average velocity (F_1, 61_ = 6.552, *p* = 0.0130), but not inner/outer ratio (F_1, 61_ = 0.003, *p* = 0.9546). There was no significant main effect of genotype on total distance traveled (F_1, 52_ = 1.748, *p* = 0.1919), total average velocity (F_1, 61_ = 0.361, *p* = 0.5493), or inner/outer ratio (F_1, 61_ = 1.840, *p* = 0.1799). There was no significant interaction between genotype and diet on total distance traveled (F_1, 52_ = 0.758, *p* = 0.3880), total average velocity (F_1, 61_ = 1.267, *p* = 0.2647), or inner/outer ratio (F_1, 61_ = 0.134, *p* = 0.7155). Within genotypes, total distance traveled was significantly decreased in HFD-fed mice relative to NC-fed mice (WT HFD vs. WT NC *p* < 0.001, Figure 2D; AD HFD vs. AD NC *p* < 0.05, Figure 2D).

### 3.3. HFD Increases Behavioral Despair in Both AD and WT Mice Compared to NC in Females

The Porsolt forced swimming test was used to measure behavioral despair. In females, diet impacted time spent immobile (F_1, 62_ = 29.77, *p* < 0.0001), but genotype did not (F_1, 62_ = 1.469, *p* = 0.2301) and there was no significant interaction between the two (F_1, 62_ = 0.235, *p* = 0.6294). In both genotypes, a HFD increased time spent immobile compared to NC-fed mice (WT HFD vs. WT NC, *p* < 0.001; AD HFD vs. AD NC, *p* < 0.01; Figure 3A). While males showed a significant main effect of diet (F_1, 49_ = 11.22, *p* = 0.0016), post hoc tests did not reveal any between-group differences. There was also no significant effect of genotype (F_1, 49_ = 1.835, *p* = 0.1818), nor an interaction between genotype and diet (F_1, 49_ = 0.043, *p* = 0.8359).

### 3.4. HFD Increases Memory Deficits in Both AD and WT Mice Compared to NC in Both Sexes

The Y-maze test was used to test short-term spatial memory. In both sexes, differences in spontaneous alternation percentage were attributed to diet (Female: F_1, 45_ = 29.25, *p* < 0.0001; Male: F_1, 42_ = 21.95, *p* < 0.0001) and genotype (Female: F_1, 45_ = 23.55, *p* < 0.0001; Male: F_1, 42_ = 22.90, *p* < 0.0001), with no significant interaction between the two (Female: F_1, 45_ = 0.395, *p* = 0.5328; Male: F_1, 42_ = 0.857, *p* = 0.3598). In both genotypes, a HFD decreased spontaneous alternation percentage compared to NC-fed mice (Female: WT NC vs. WT HFD, *p* < 0.001; AD NC vs. AD HFD, *p* < 0.01; Figure 4A. Male: WT NC vs. WT HFD, *p* < 0.01; AD NC vs. AD HFD, *p* < 0.05; Figure 4B). However, in both sexes, AD mice showed reduced overall short-term memory performance, with NC-fed AD mice performing as poorly as did the HFD-fed WT mice, and HFD-fed AD mice performing significantly worse than all other groups (Male and Female AD HFD vs. WT HFD, *p* < 0.05; Figure 4).

The Barnes maze task was used to test spatial memory. Learning was apparent in all groups (Figure 5A,D). Differences in learning started on day two of the test (Females: WT NC vs. WT HFD, *p* < 0.05; WT NC vs. AD NC, *p* < 0.01; WT NC vs. AD HFD, *p* < 0.01. Males: WT NC vs. AD HFD, *p* < 0.05.), and continued through days three (Females: WT NC vs. WT HFD, *p* < 0.01; WT NC vs. AD NC, *p* < 0.001; WT NC vs. AD HFD, *p* < 0.001; WT HFD vs. AD NC, *p* < 0.05. Males: WT NC vs. WT HFD, *p* < 0.05; WT NC vs. AD NC, *p* < 0.01, WT NC vs. AD HFD, *p* < 0.0001.) and four (Females: WT NC vs. WT HFD, *p* < 0.0001; WT NC vs. AD NC, *p* < 0.0001; WT NC vs. AD HFD, *p* < 0.0001; WT HFD vs. AD NC, *p* < 0.01; WT HFD vs. AD HFD, *p* < 0.001; AD NC vs. AD HFD, *p* < 0.05. Males: WT NC vs. WT HFD, *p* < 0.01; WT NC vs. AD NC, *p* < 0.001; WT NC vs. AD HFD, *p* < 0.0001; AD NC vs. AD HFD, *p* < 0.05.). In the probe test, differences in time spent in the target quadrant were attributed to diet (Female: F_1, 45_ = 39.06, *p* < 0.0001; Male: F_1, 57_ = 35.34, *p* < 0.0001) and genotype (Female: F_1, 45_ = 20.11, *p* < 0.0001; Male: F_1, 57_ = 14.85, *p* = 0.0003), with no significant interaction between the two (Female: F_1, 45_ = 0.758, *p* = 0.3887 Male: F_1, 57_ = 0.054, *p* = 0.8166) in both sexes. As with short-term spatial memory, post hoc tests showed that while AD mice fed NC had impaired performance relative to NC-fed WT mice (Females: *p* < 0.01, Figure 5B; Males: *p* < 0.05, Figure 5E), a HFD significantly exacerbated performance deficits in both genotypes (Females: WT NC vs. WT HFD, *p* < 0.0001; AD NC vs. AD HFD, *p* < 0.01; Figure 5B,C. Males: WT NC vs. WT HFD, *p* < 0.001; AD NC vs. AD HFD, *p* < 0.01; Figure 5E,F).

### 3.5. HFD Increases Microgliosis in the Hippocampus, Cortex, and Hypothalamus of AD and WT Mice in Both Sexes

Microgliosis was measured by immunohistochemical staining for IBA1 in brain sections. In females, IBA expression in AD mice was higher in all brain regions examined (Figure 6). In the hippocampus, cortex, ventromedial hypothalamus (VMH), and arcuate nucleus (ARC), IBA expression showed significant main effects of diet (Hippocampus F_1, 14_ = 20.55, *p* = 0.0005; Cortex F_1, 15_ = 20.44, *p* = 0.0004; VMH F_1, 15_ = 27.68, *p* < 0.0001; ARC F_1, 15_ = 25.59, *p* = 0.0001) and genotype (Hippocampus F_1, 14_ = 57.49, *p* < 0.0001; Cortex F_1, 15_ = 36.20, *p* < 0.0001; VMH F_1, 15_ = 15.22, *p* = 0.0014; ARC F_1, 15_ = 14.34, *p* = 0.0018), with no significant interaction between the two (Hippocampus F_1, 14_ = 0.087, *p* = 0.7717; Cortex F_1, 15_ = 0.010, *p* = 0.9222; VMH F_1, 15_ = 0.541, *p* = 0.4735; ARC F_1, 15_ = 0.773, *p* = 0.3932). In all regions, a HFD increased IBA expression regardless of genotype (*p* < 0.01–0.05, Figure 7A,C,E,G). However, AD animals expressed higher baseline levels of IBA staining, with NC-fed AD mice significantly (hippocampus, *p* < 0.001; cortex, *p* < 0.01) or somewhat higher (VMH, ARC) than NC-fed WT controls.

In males, IBA expression showed a similar pattern (Figure 7). In the hippocampus, cortex, and VMH, IBA expression showed significant main effects of diet (Hippocampus F_1, 11_ = 25.94, *p* = 0.0003; Cortex F_1, 12_ = 35.93, *p* < 0.0001; VMH F_1, 12_ = 27.48, *p* = 0.0002) and genotype (Hippocampus F_1, 11_ = 39.66, *p* < 0.0001; Cortex F_1, 12_ = 54.18, *p* < 0.0001; VMH F_1, 12_ = 38.65, *p* < 0.0001), with no significant interaction between the two (Hippocampus F_1, 11_ = 0.001, *p* = 0.9728; Cortex F_1, 12_ = 1.298, *p* = 0.2769; VMH F_1, 12_ = 0.302, *p* = 0.5925; ARC F_1, 15_ = 0.105, *p* = 0.7516). As in females, NC-fed AD mice showed higher baseline IBA than did NC-fed WT (*p* < 0.01 in all regions), with significant exacerbation of IBA in all three regions under HFD in both genotypes (*p* < 0.01–0.05; Figure 7A,C,E). Unlike female mice, no differences were observed in ARC (Figure 7G).

### 3.6. HFD Increases Aβ_1–42_ Deposits in the Hippocampus and Cortex of AD in Both Sexes

Immunohistochemistry for Aβ12F4 was performed on brain sections to measure Aβ_1–42_ deposits. In both males and females, HFD-fed AD mice had increased Aβ12F4 staining compared to NC-fed AD mice in the hippocampus (Female: *p* < 0.05, Figure 8A,B; Male: *p* < 0.01, Figure 8E,F) and cortex (Female: *p* < 0.05, Figure 8C,D; Male: *p* < 0.01, Figure 8G,H).

### 3.7. HFD Increases Aβ_17–24_ Deposits in the Hippocampus and Cortex of AD in Both Sexes

Immunohistochemistry for Aβ4G8 was performed on brain sections to measure Aβ_17–24_ deposits. In both males and females, HFD-fed AD mice had increased hippocampal (Female: *p* < 0.05, Figure 9A,B; Male: *p* < 0.05, Figure 9E,F) and cortical Aβ4G8 (Female: *p* < 0.05, Figure 9C,D; Male: *p* < 0.05, Figure 9G,H) compared to NC-fed AD mice.

## 4. Discussion

AD is the leading cause of dementia and is prevalent worldwide [29]. As obesity is often linked with an earlier onset of AD and increased AD neuropathology [1,2], and diets high in saturated fatty acids are known to potentiate neuroinflammatory diseases such as AD [6,7,8], we investigated the effects of a HFD on cognition, neuroinflammation, and AD pathology in a young adult AD mouse model. HFD-fed mice had a larger increase in fat mass and lean mass compared to their NC-fed counterparts. There was no difference between genotypes for changes in fat mass and lean mass (Figure 1). We also measured non-fasted blood glucose levels before termination. The results showed an increase in blood glucose in HFD groups, but no differences were seen between genotypes (Appendix A). Through an open field test, we analyzed the locomotor activity and anxiety-like behavior of animal groups. The results showed that HFD-fed mice had decreased locomotor activity as measured by total distance traveled and average velocity in females, and total distance traveled in males (Figure 2A,B,D). This was coupled with an increase in anxiety-like behavior in female HFD groups, as revealed in an increase in time spent in the periphery of the arena (Figure 2C). These results are consistent with other HFD studies showing decreased locomotor activity and increased anxiety-like behavior [30,31]. Through the Porsolt forced swimming test, we examined the differences of behavioral despair displayed by our treatment groups. The results showed that HFD-fed female mice displayed increased behavioral despair compared to NC-fed mice. No differences in behavioral despair were seen between genotypes (Figure 3). The increase in behavioral despair in female HFD groups has been seen in other HFD studies of depression-like behavior [32]. As no differences were seen between genotypes in HFD groups in body composition, blood glucose, locomotor activity, anxiety-like behavior, or behavioral despair, differences observed in spatial memory, neuroinflammation, and AD pathology are not likely to be attributed to these variables.

We tested the spatial memory of treatment groups using Y-maze and Barnes maze tests. Y-maze results in both sexes showed that a HFD led to decreased spontaneous alternation percentages in both genotypes. HFD-fed AD mice also showed decreased spontaneous alternation percentages when compared to HFD-fed WT mice (Figure 4). As the Y-maze is a test of short-term spatial working memory [33], our results show that a HFD leads to impaired short-term spatial working memory in both AD and WT mice. The results in WT mice are consistent with our previous findings [9]. Our results also show that HFD-fed AD mice displayed the largest impairment of short-term spatial working memory compared to any other group. This is consistent with work conducted by others that have shown short-term cognitive deficits in HFD-fed mouse models of AD [34]. To test learning and spatial reference memory, the Barnes maze was utilized. The results showed that a HFD led to decreased time spent in the target quadrant after 4 days of testing in both genotypes. In females, HFD-fed AD mice also showed decreased time spent in the target quadrant when compared to HFD-fed WT mice. HFD-fed AD mice also displayed more deficits in learning compared to any other group (Figure 5A). The results in WT mice were consistent with our previous work [9]. Our results show that AD mice performed worse on this task compared to their WT counterparts; this is also consistent with the work conducted by others using transgenic AD models [35]. Overall, the current results show that genotype and diet both effect performance in memory tasks. Mice fed a HFD performed worse than their NC counterparts. AD mice also performed worse than their WT counterparts. HFD-fed AD mice exhibited the largest cognitive deficits out of all groups. These behavioral results support the idea that a HFD exacerbates AD-related cognitive decline.

To examine the effect of a HFD on neuroinflammation, immunohistochemistry was performed for IBA1. The results for IBA1 staining showed increased microgliosis in HFD and AD groups for both sexes (Figure 6 and Figure 7). In females, HFD-fed AD mice had the largest amount of IBA1 staining in the hippocampus, cortex, ventromedial hypothalamus, and arcuate nucleus (Figure 6). The same was seen in males in the hippocampus, cortex, and ventromedial hypothalamus (Figure 7). Others have shown that a HFD leads to increased inflammation in the arcuate nucleus and ventromedial hypothalamus [36,37]. Hypothalamic inflammation has been known to contribute to obesity and diabetes-related neurodegeneration [38]. We and others have also previously shown that a HFD leads to increased hippocampal neuroinflammation and microgliosis [9,39]. Inflammation in the hippocampus and cortex has been strongly associated with Aβ plaque deposition in AD [11]. Microglia can bind Aβ fibrils via cell surface receptors [40]. While this can assist in the clearance of Aβ, sustained microglial activation can lead to the release of proinflammatory cytokines, including TNF-α, IL-1, IL-6, IL-12, and IL-18 [41]. In turn, exposure to these cytokines leads to the downregulation of genes with roles in Aβ clearance [42]. This can lead to the progression of Aβ pathology and neurodegeneration [41]. Immunohistochemistry was also performed for Aβ12F4 (Aβ_1–42_ deposits), and Aβ4G8 (Aβ_17–24_ deposits). The results for Aβ12F4 showed increased Aβ_1–42_ deposits in HFD-fed AD mice compared to NC-fed AD mice. This was seen in the hippocampus and cortex regions in both sexes (Figure 8). The same differences were observed for Aβ4G8, showing increased Aβ_17–24_ deposits in HFD-fed AD mice (Figure 9). Aβ deposition is a hallmark of AD pathology. Aβ is formed via the proteolysis of amyloid precursor protein (APP) by presenilin-1 (PS1) of the γ-secretase complex [43]. Aβ monomers can aggregate into oligomers and fibrils, the latter of which can further aggregate to the point of becoming plaques [44]. This is considered an early initiating event of AD pathogenesis that eventually leads to memory impairment and cognitive decline [45]. As the hippocampus is an important region for learning and memory, hippocampal Aβ deposition has been shown to impair long-term potentiation [46]. Aβ deposition has also been found in the cerebral cortex of AD patients [47], a region also involved with spatial memory [48]. Aβ_1–42_ deposits in particular are known to make up most of the Aβ plaques found in the AD brain [49]. As such, our results show that young adult AD mice fed a HFD experience increased Aβ_1–42_ and overall Aβ deposits in multiple brain regions.

Our present results provide evidence that a HFD exacerbates neuroinflammation and Aβ pathology in a young adult AD mouse model, leading to increased memory deficits and cognitive decline in both sexes. A HFD leads to an increase in microglia activation, contributing to an increase in proinflammatory cytokine signaling. This activation of microglia can lead to a decrease in Aβ uptake and clearance, leading to increased Aβ plaques. This contributes to the initiation of the cascade that eventually leads to memory impairment and cognitive decline. Early adulthood obesity has been correlated with late-life dementia independent of mid- and late-life obesity [3], and our results provide further evidence of this relationship. Further, we found that this relationship is present in both males and females despite their different risk profiles. A lot of work is still needed to be carried out to determine the role of microglia in HFD-induced cognitive decline and neurodegeneration. Recent advances in transcriptomic technologies have helped in identifying microglial heterogeneity in the AD brain [50]. Utilizing spatial phenotyping will help to elucidate the microglia-specific mechanisms that lead to increased neuroinflammation and Aβ deposition in various brain regions due to high saturated fatty acid intake. These studies will provide further clarity on the links between obesity and Alzheimer’s disease.

## Figures and Tables

**Figure 1 nutrients-15-02494-f001:**
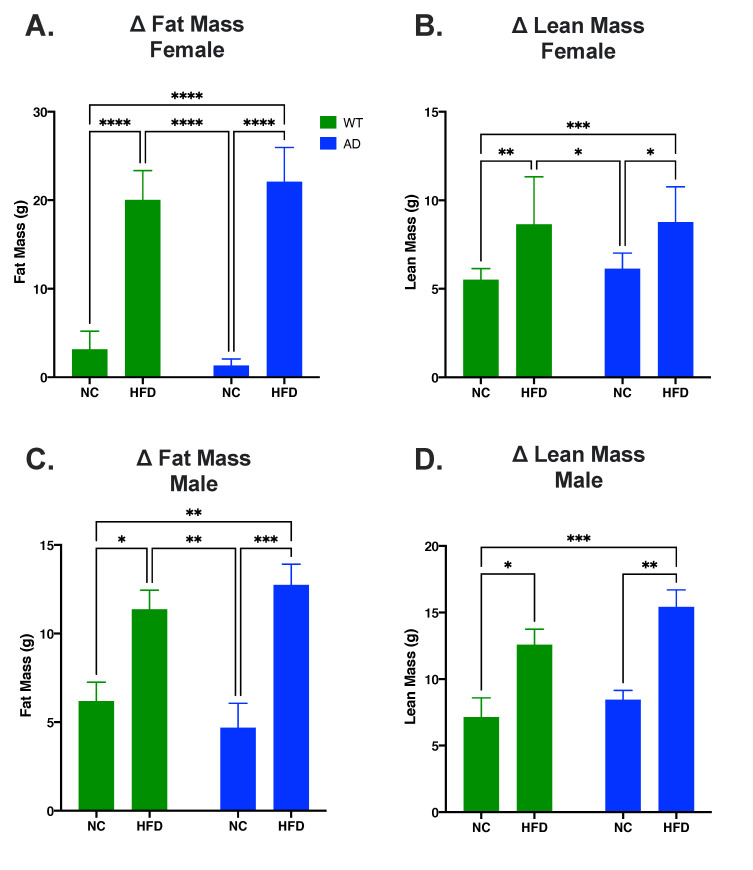
HFD increases fat mass and lean mass in both AD and WT mice. (**A**). Change in fat mass in female mice. (**B**). Change in lean mass in female mice. (**C**). Change in fat mass in male mice. (**D**). Change in lean mass in male mice. Data analyzed via two-way ANOVA and Holm–Sidak’s multiple comparisons test. n = 10–14. * *p* < 0.05, ** *p* < 0.01, *** *p* < 0.001, **** *p* < 0.0001.

**Figure 2 nutrients-15-02494-f002:**
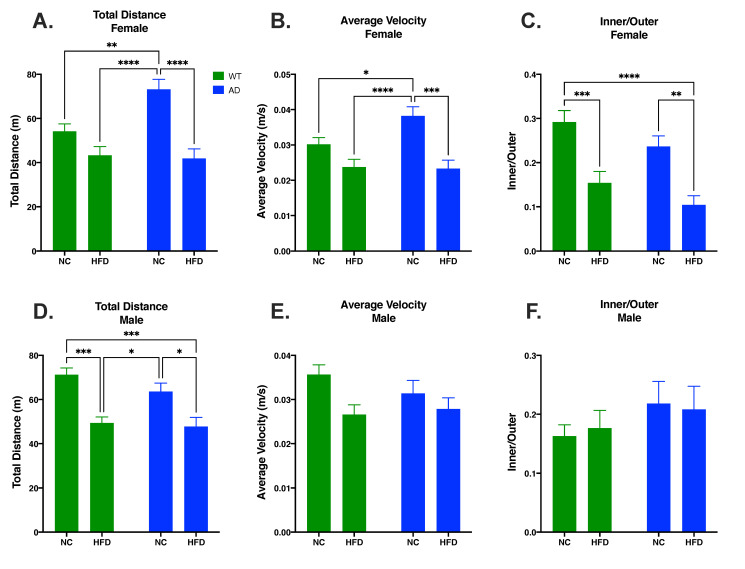
HFD decreased distance traveled (**A**) and velocity (**B**) and increased anxiety-like behavior (**C**) in female mice. In male mice, HFD impacted total distance travelled (**D**) but not average velocity (**E**) or anxiety-like behavior (**F**). Data analyzed via two-way ANOVA and Holm–Sidak’s multiple comparisons test. n = 12–21. * *p* < 0.05, ** *p* < 0.01, *** *p* < 0.001, **** *p* < 0.0001.

**Figure 3 nutrients-15-02494-f003:**
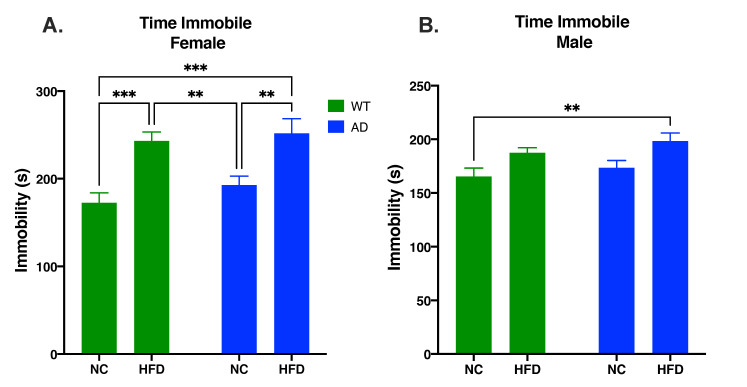
HFD increases behavioral despair in female (**A**) and male mice (**B**). Data analyzed via two-way ANOVA and Holm–Sidak’s multiple comparisons test. n = 13–18. ** *p* < 0.01, *** *p* < 0.001.

**Figure 4 nutrients-15-02494-f004:**
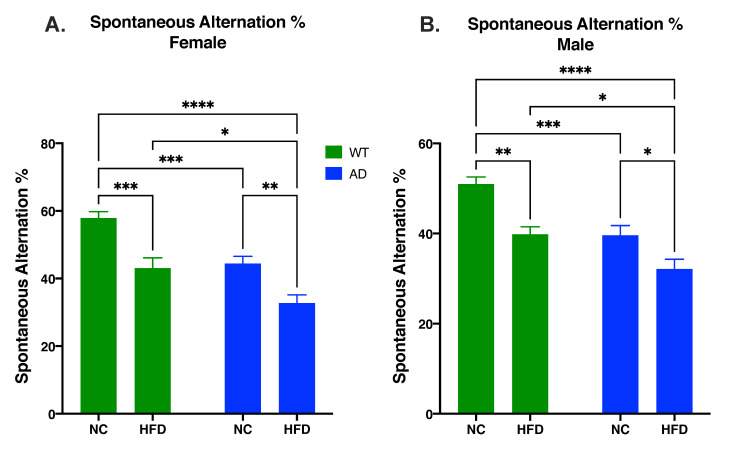
HFD increases memory deficits as measured by spontaneous alternation percentage in both female (**A**) and male mice (**B**). Data analyzed via two-way ANOVA and Holm–Sidak’s multiple comparisons test. n = 10–14. * *p* < 0.05, ** *p* < 0.01, *** *p* < 0.001, **** *p* < 0.0001.

**Figure 5 nutrients-15-02494-f005:**
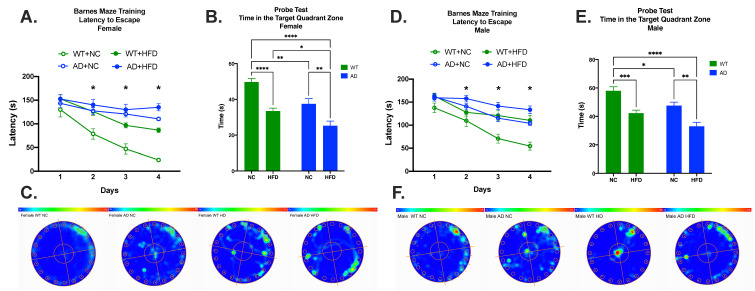
HFD increases memory deficits in both sexes. Female mice: Latency to escape (**A**), time in the target quadrant (**B**), and heatmap (**C**). Male mice: Latency to escape (**D**), time in the target quadrant (**E**), and heatmap (**F**). Data analyzed via two-way ANOVA and Holm–Sidak’s multiple comparisons test. n = 11–17. * *p* < 0.05, ** *p* < 0.01, *** *p* < 0.001, **** *p* < 0.0001.

**Figure 6 nutrients-15-02494-f006:**
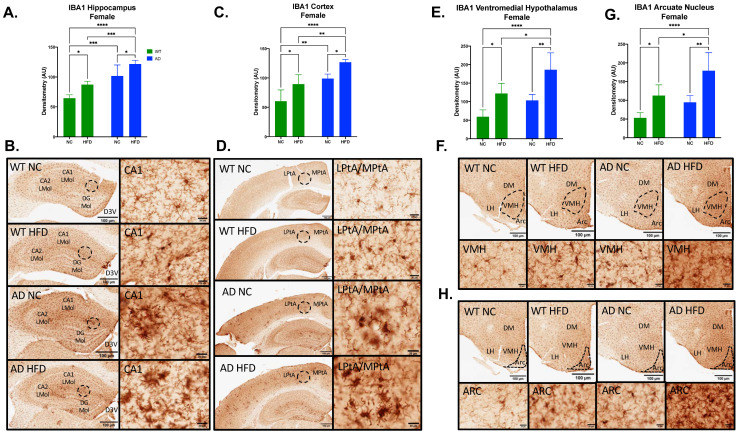
HFD increases IBA1 expression in the hippocampus, cortex, ventromedial hypothalamus (VMH), and arcuate nucleus (ARC) in female mice. Densitometry analysis (**A**) and immunohistochemical staining (**B**) of IBA1 in the hippocampus, cortex (**C**,**D**), VMH (**E**,**F**), and ARC (**G**,**H**). Sections are 40 μm thick. Scale bars are 100 μm and 20 μm for inset. Brain regions are labeled CA1 LMol: CA1 lacunosum moleculare layer; CA2 LMol: CA2 lacunosum moleculare layer; DG Mol: dentate gyrus molecular layer; D3V: third ventricle; LPtA: lateral parietal association cortex; MPtA: medial parietal association cortex; DM: dorsomedial nucleus; LH: lateral hypothalamic area; VMH: ventromedial hypothalamus; ARC: arcuate nucleus. Data analyzed via two-way ANOVA and Holm–Sidak’s multiple comparisons test. n = 4–5. * *p* < 0.05, ** *p* < 0.01, *** *p* < 0.001, **** *p* < 0.0001.

**Figure 7 nutrients-15-02494-f007:**
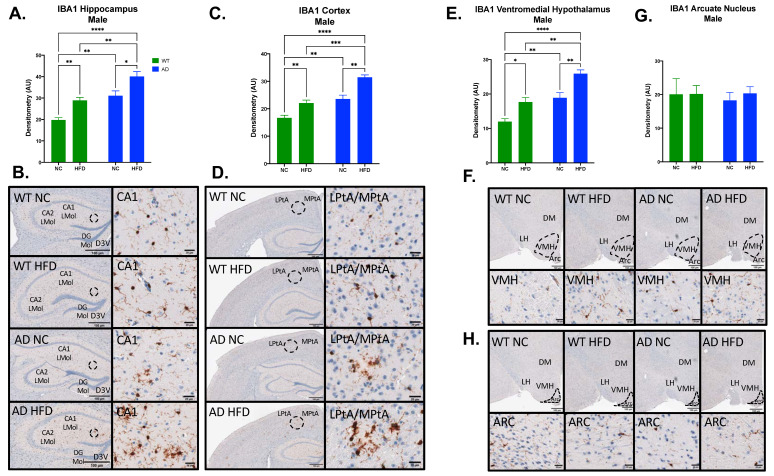
HFD increases IBA1 expression in the hippocampus, cortex, and ventromedial hypothalamus (VMH), but not in the arcuate nucleus (ARC) in male mice. Densitometry analysis (**A**) and immunohistochemical staining (**B**) of IBA1 in the hippocampus, cortex (**C**,**D**), VMH (**E**,**F**) and ARC (**G**,**H**). Sections are 5 μm thick. Scale bars are 100 μm and 20 μm for inset. Brain regions are labeled CA1 LMol: CA1 lacunosum moleculare layer; CA2 LMol: CA2 lacunosum moleculare layer; DG Mol: dentate gyrus molecular layer; D3V: third ventricle; LPtA: lateral parietal association cortex; MPtA: medial parietal association cortex; DM: dorsomedial nucleus; LH: lateral hypothalamic area; VMH: ventromedial hypothalamus; ARC: arcuate nucleus. Data analyzed via two-way ANOVA and Holm–Sidak’s multiple comparisons test. n = 4–5. * *p* < 0.05, ** *p* < 0.01, *** *p* < 0.001, **** *p* < 0.0001.

**Figure 8 nutrients-15-02494-f008:**
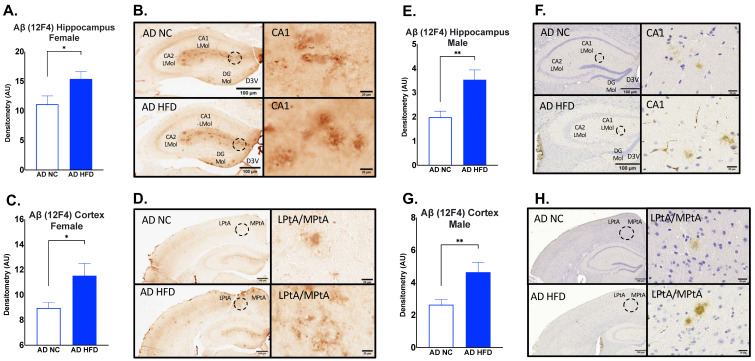
HFD increases Aβ_1–42_ deposits in the hippocampus and cortex in both sexes. Densitometry analysis and immunohistochemical staining of Aβ12F4 in the hippocampus (**A**,**B**) and cortex (**C**,**D**) of female mice, and in the hippocampus (**E**,**F**) and cortex (**G**,**H**) of male mice. Sections are 40 μm thick for females and 5 μm for males. Scale bars are 100 μm and 20 μm for inset. Brain regions are labeled CA1 LMol: CA1 lacunosum moleculare layer; CA2 Lmol: CA2 lacunosum moleculare layer; DG Mol: dentate gyrus molecular layer; D3V: third ventricle; LptA: lateral parietal association cortex; MptA: medial parietal association cortex. Data analyzed via t-test. N = 4–5. * *p* < 0.05, ** *p* < 0.01.

**Figure 9 nutrients-15-02494-f009:**
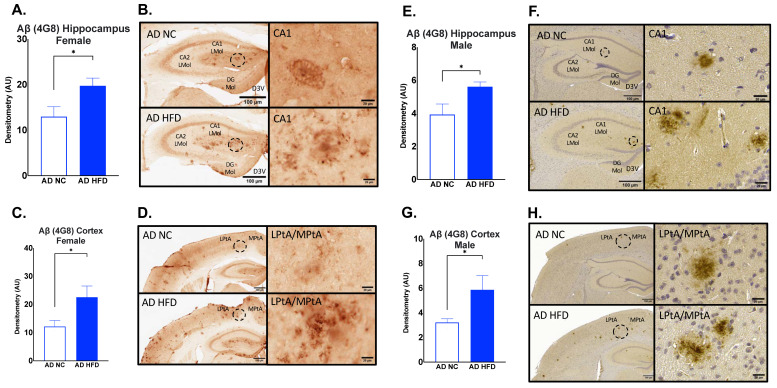
HFD increases Aβ_17–24_ deposits in the hippocampus and cortex in both sexes. Densitometry analysis and immunohistochemical staining of Aβ4G8 in the hippocampus (**A**,**B**) and cortex (**C**,**D**) of female mice, and in the hippocampus (**E**,**F**) and cortex (**G**,**H**) of male mice. Sections are 40 μm thick for females and 5 μm for males. Scale bars are 100 μm and 20 μm for inset. Brain regions are labeled CA1 LMol: CA1 lacunosum moleculare layer; CA2 LMol: CA2 lacunosum moleculare layer; DG Mol: dentate gyrus molecular layer; D3V: third ventricle; LPtA: lateral parietal association cortex; MPtA: medial parietal association cortex. Data analyzed via t-test. n = 4–5. * *p* < 0.05.

## Data Availability

Data sets generated during this study are the property of the U.S. Department of Veterans Affairs and will be made available upon request.

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
