# Peer review of "Early Life Obesity Increases Neuroinflammation, Amyloid Beta Deposition, and Cognitive Decline in a Mouse Model of Alzheimer’s Disease"

_nutrients, 2023, doi:10.3390/nu15112494_

Round 1
Reviewer 1 Report
The authors hypothesized that proinflammatory activation of brain microglia in obesity would exacerbate Alzheimer’s disease (AD) pathology and increase accumulation of amyloid beta (Aβ) plaques. The authors tested cognitive function in 8-month-old male and female APP/PS1 transgenic mice fed with high fat diet (HFD) which started at 1.5 months of age. Locomotor activity, anxiety-like behavior, behavioral despair, and spatial memory were all assessed through behavioral tests. Microgliosis and Aβ deposition was measured in multiple brain regions through immunohistochemical analysis. The results showed that HFD decreased locomotor activity, while increasing anxiety-like behavior and behavioral despair independent of genotype. HFD led to increased memory deficits in both sexes, with HFD-fed APP/PS1 mice performing the worst out of all groups. Immunohistochemical analysis showed increased microgliosis in HFD-fed mice. This was accompanied by an increase in Aβ deposition in the HFD-fed APP/PS1 mice. Together, the results supported that HFD-induced obesity exacerbated neuroinflammation and Aβ deposition in a young adult AD mouse model, leading to increased memory deficits and cognitive decline in both sexes.
The manuscript is interesting, but it also contains several problems to be resolved.
(Major points)
1) Figure 2; the HFD-fed AD mice showed decreased traveling distances and velocities, but is it possible simply that “the mice with heavier weight showed less locomotive activity or more fatigue”, independent of cognitive function or emotion?
The authors should prove that all the performance in the behavioral tests were irrelevant to simple weight gain, too.
2) Figure 2A shows longer total traveling distance in NC (normal chow)-fed female AD mice but did the authors think it meant higher anxiety? HFD-fed female AD mice showed shorter traveling distance but does it mean HFD-fed female AD mice had less anxiety?
Figure 2D showed NC-fed AD mice had no difference in total traveling distance, so does it mean NC-fed male AD mice had no increased anxiety? Furthermore, Figure 2D does not seem to show “diet” × “genotype” interaction, and it would not support the authors’ hypothesis.
The authors should explain the meanings of an open field test in this study more precisely.
3) Figure 3A showed HFD-fed wild type (WT) female mice spent more time to be immobile, but does it mean the level of “despair” or “struggling because of over-weight”? The authors also mentioned “No differences of behavioral despair were seen between genotypes (Figure 3)” in the Discussion (Page 11, lines 334-335) but it should be mentioned in the Result (3.3.) with appropriate statistical data. And if so, the result would not support the authors’ hypothesis.
4) The authors should show statistical results more precisely and explicitly, especially for the interaction of confounding factors with ANOVA.
Figures 1, 3, 4, 5, 6 and 7; the interaction between “diet” and “genotype”.
Figure 2; the interaction of “diet” × “genotype” × “sex”.
5) Figures 1-5 implied the differences between the groups, but the graphs lack asterisks to indicate statistical significance.
6) The line graphs in Figure 5 suggest inter-group differences, but the results of statistical tests are not indicated.
7) The authors mentioned “There was no difference between genotypes for changes in fat mass, lean mass, or body weight (Figure 1)” in the Discussion (Page 11, lines 321-323) but it should be mentioned in the Result (3.1.) with appropriate statistical data.
(Minor points)
1) Page 1-line 31 and page 11-line 317: “AD neuropathy” should be “AD neuropathology” because “neuropathy” means “disease(s) of peripheral nerve(s)”.
2) All abbreviations should be specified before use.
Page 1, line 32; BMI, body mass index
Page 2, line 70; Tg, transgene (or transgenic)
Page 2, line 72; qRT-PCR, quantitative real-time polymerase chain reaction
Page 3, line 135; PBS, phosphate-buffered saline
Page 4, line 153; DAB, 3,3'-diaminobenzidine
Figures 6 and 7; “LPtA/MPtA”, (Please specify it.); DG, dentate gyrus; “LMol”, (Please specify it.)
Reviewer 2 Report
In the manuscript submitted by So et. al. titled as ‘Early life obesity increases neuroinflammation, amyloid beta deposition, and cognitive decline in a mouse model of Alzheimer’s disease’, the authors found high fat diet (HFD) exacerbate cognitive impairment, microgliosis and amyloid pathology in APP/PS1 mice. They included both non-amyloid- amyloid-bearing mice and included both sexes.
However, the big concern of this paper is its novelty given that there are several other papers investigated whether and how HFD affects amyloid pathologies, such as PMID: 18926603, PMID: 34864680, PMID: 34212829, PMID: 34245097 and others. Specifically, in PMID: 34864680 which is not cited by So et.al, they used similar paradigm of the HFD treatment from 2.5 to 6.5 months of age while So et.al conducted the HFD from 1.5 to 7.5 months of age with the same mouse model. Unfortunately, not only the design, but also the conclusion themselves were lacking novelty even though their results support their conclusions well. If the authors could do more mechanism investigations like what molecular signature of the microglia responding to HFD in the absence or presence of Aβ, it will definitely make the paper comprehensive.
Additionally, there is another concern. The references cited the authors seem to not match to the text. Taking the first 8 references as examples, the references are non-relevant with their statements in the introduction. Also, refence 4 and 19 are the same paper.
Two minor comments here:
1. Fig. 1-5, please annotate the significance levels in the figures as in Fig. 6-9.
2. Line 302, “of” is lacked.
Good. No problem.
Round 2
Reviewer 1 Report
(Please see the attached file.)

Reviewer 2 Report
I appreciate the feedback from the authors. The introduction has been improved by citing more HFD related studies using AD mouse models and clarification of the differences compared with their current study. The reference issue has been corrected. Others have been addressed sufficiently.
Author Response
We thank the reviewer for their comments and suggestions.